# Task-Driven Learning of Contour Integration Responses in a V1 Model

**Salman Khan**
Centre for Theoretical Neuroscience
University of Waterloo
Waterloo, Ontario, Canada
s362khan@uwaterloo.ca

**Alexander Wong**
Vision and Image Procession(VIP) Group
University of Waterloo
Waterloo, Ontario, Canada
alexander.wong@uwaterloo.ca

**Bryan Tripp**
Centre for Theoretical Neuroscience
University of Waterloo
Waterloo, Ontario, Canada
bptripp@uwaterloo.ca

## Abstract

Under difficult viewing conditions, the brain's visual system uses a variety of modulatory techniques to augment its core feed-forward signals. Incorporating these into artificial neural networks can potentially improve their robustness. However, before such mechanisms can be recommended, they need to be fully understood. Here, we present a biologically plausible model of one such mechanism, contour integration, embedded in a task-driven artificial neural network. The model is neuroanatomically grounded and all its connections can be mapped onto existing connections in the V1 cortex. We find that the model learns to integrate contours from high-level tasks including those involving natural images. Trained models exhibited several observed neurophysiological and behavioral properties. In contrast, a parameter matched feed-forward control achieved comparable task-level performance but was largely inconsistent with neurophysiological data.

## 1 Introduction

Contour integration [1, 2, 3, 4] is a phenomenon in the V1 cortex where stimuli from outside a neuron's classical receptive field (cRF) modulate its feed-forward responses. In particular, responses are enhanced if a preferred stimulus within the cRF is part of a larger contour. Under difficult viewing conditions, the visual ventral stream uses contour integration to *pop out* smooth natural contours (see Figure 2). Contour integration is thought to be mediated by intra-area lateral and higher-layer feedback connections. Past computational models [5, 6] have tested potential mechanisms in isolation and replicated observed neurophysiological data. However, apart from some recent work [7, 8], past studies have done little to explore the role of contour integration in the perception of naturalistic scenes.

Separately, recent advances in deep neural networks have surpassed human-level performance on several high-level vision tasks such as object classification [9, 10, 11] and object segmentation [12, 13]. This enables the possibility of embedding a contour integration model within a task-oriented system to quantitatively analyze the relationships between low-level contour integration and high-level behavioral tasks. However, neural networks are notorious black box function approximators that learn complex mapping between inputs and outputs by whatever mechanism minimizes a global cost function. To fairly compare decision-making strategies and draw robust conclusions, architectures needed to be carefully considered and models need to be analyzed at multiple levels [14].

2nd Workshop on Shared Visual Representations in Human and Machine Intelligence (SVRHM), NeurIPS 2020.

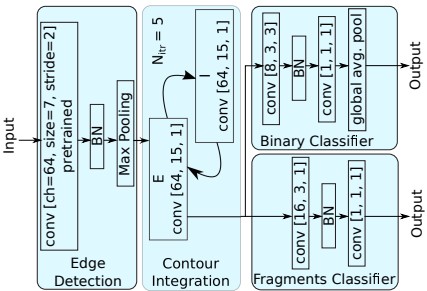

Figure 1: Model architecture. The square brackets specify the number of kernels, kernel size, and stride length for each convolutional layer.

In this work, we explore whether contour integration can be learnt from high-level visual tasks and whether learnt mechanisms are consistent with observed neurophysiological and behavioral data. Li et. al. [2] concurrently monitored behavioral performance and V1 neural responses of macaque monkeys. At the behavioral level, contours became more salient as lengths increased. When contours extended in the direction of the preferred orientation of V1 neurons, their firing rate monotonically increased. Conversely, when spacing between fragments increased, contours became less salient and V1 firing rates decreased monotonically. In a similar manner, we analyzed trained models behaviorally at the prediction level and neurophysiologically at the output of the embedded contour integration model.

**Related work**   Recently, recurrent neural network (RNN) models of intra-layer horizontal connections have been proposed [7, 8, 15]. Similar to our model, these models learn lateral connectivity patterns from high-level tasks. However, different from our model, they use complex multi-parameter gates inspired from Gated Recurrent Unit (GRU) [16] and Long Short Term Memory (LSTM) [17] RNNs. These complex multi-parameter gates are functions of inputs, outputs and the internal states of multiple neurons and establish multiple parallel paths between them. Some of these connections may not exist in the brain and their complex connectivity patterns makes it difficult to map those that do exist back to the neuroanatomy. In contrast, our simpler model is more biologically aligned and all its connections can be directly mapped onto existing V1 connections. As pointed out by Fenke et. al. [14], conclusions generalized beyond tested architectures can be fragile. Furthermore, we focus primarily on replicating the brain's mechanism of contour integration rather than achieving the best performance on high-level tasks using recurrence.

## 2   Model

The central component of the model is the contour integration (CI) layer. It models V1 orientation columns and the interactions between them. Each orientation column represents a population of neurons that respond to edges of similar orientations at a particular spatial location. Each orientation column is modeled with a pair of nodes. The interaction between nodes is derived from the differential equation model of Piech et. al. [6], in which excitatory (E) and inhibitory (I) nodes connect locally with each other and selectively with nodes in neighboring columns. To incorporate this structure into neural networks, we use Euler's method to express its components as difference equations and use RNN methods to make them easier to train [7, 18]. While the recurrent architecture is borrowed from [6], we have not constrained the signs of the weights, although this is a topic of future work. The final form of these interactions is represented by

$$x_t = (1 - \sigma(a))x_{t-1} + \sigma(a)\left[-J_{xy}f_y(y_{t-1}) + I_{0e} + I + \text{Relu}(W_e \circledast F_x(x_{t-1}))\right], \quad (1)$$

$$y_t = (1 - \sigma(b))y_{t-1} + \sigma(b)\left[J_{yx}f_x(x_t) + I_{0i} + \text{Relu}(W_i \circledast F_x(x_t))\right]. \quad (2)$$

Here, $x$ and $y$ are the membrane potential of E and I nodes, respectively, $f_.(.)$ is a non-linear activation function that transforms membrane potentials into firing rates, $1/\tau_x = \sigma(a)$, $1/\tau_y = \sigma(b)$ are membrane time constants, $J_{xy}, J_{yx}$ are local I $\rightarrow$ E, E $\rightarrow$ I connection strengths, $W_e$ are lateral connections from E nodes in nearby columns to E, $W_i$ are lateral connections from nearby E nodes to I, $F_.(.)$ is the output of all modeled nodes, $\circledast$ is the convolution operator, $\sigma()$ is the Sigmoid function which keeps time constants positive, $I_{0.}$ is a node's background activity, and $I$ is the external input. E nodes also locally self connect, E $\rightarrow$ E. This is included in $W_e$ which connects neighboring columns at the same spatial location as well. Importantly, while we borrow the network structure of [6], we do not constrain the signs of the weights, meaning that the excitatory and inhibitory roles are not enforced. The direct influence of $x_{t-1}$ on $x$ has a similar form to the indirect influence via $y_{t-1}$, suggesting that similar computations may be possible with or without such constraints. But we found

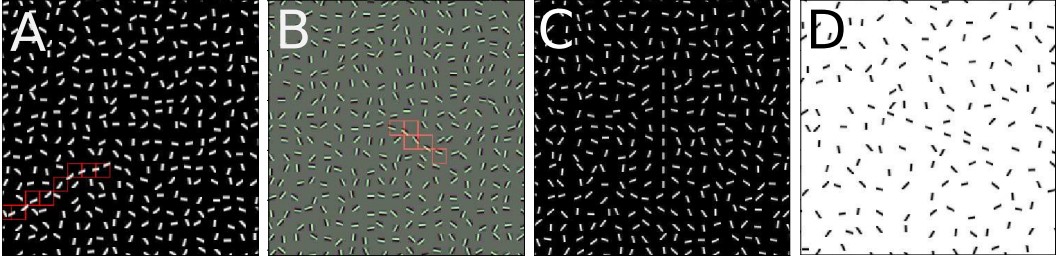

Figure 2: Contour fragments stimuli. A and B, Training stimuli. All fragments are identical Gabors. The orientation of a few adjacent fragments were aligned to form a smooth contour (highlighted in red). Contours differed in their location, orientation, length, inter-fragment curvature and their component Gabors. C and D, Test stimuli use to analyze the impact of length and inter-fragment spacing.

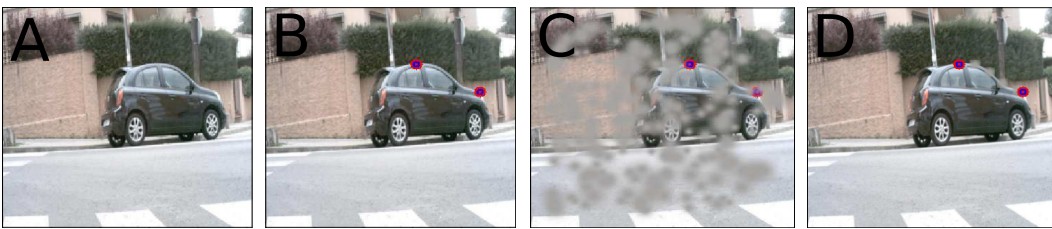

Figure 3: Contour tracing in natural images stimuli. A, Sample image from the BIPED dataset [20]. B, Using its edge label two markers are randomly placed on edge pixels. C, In training, images are punctured with occlusion bubbles to randomly fragment contours. D, To analyze the impact of fragment spacing, bubbles are placed along contours to fragment them with different spacing.

that constraining the weights greatly slows learning. We leave exploration of this issue for future work.

Lateral connections of V1 orientation columns are sparse and preferentially connect with neighbors with similar orientations, [19]. Existing models, [6, 5], use a fixed association-field [1] connection structure for each orientation column. In our implementation, we do not force a fixed lateral structure and learn it through training. All E-I pairs are connected over a defined spatial area and a sparsity constraint retains only the most prominent connections. All local connections and time constants are similarly learnt. The CI layer sits on-top an edge extraction layer (shallow layer of a neural network). E-I pairs are defined for each input channel and spatial location. Parameters are shared across spatial location but not across channels. Feed-forward input is received only by E nodes. After iterating through the CI layer for $N_{itr}$ steps, E node outputs are passed to the next layer. The spatial extent of V1 lateral connections $S$ is up to 8 times the cRF of V1 neurons [19]. Consistent with this, we define $S$ to be much larger than edge extracting kernels. For edge extraction we use the first convolutional layer of a ResNet50 [9] that was pre-trained on ImageNet. We added batch normalization and max pooling layers before the CI layer. Not only did this reduce computational complexity but it always improved performance as well. The CI layer's output was passed to classification blocks that mapped activations to desired outputs. The capacities of these blocks were kept to a minimum to allow the CI layer to do most of the work. Separate classification blocks were used for the contour fragment detection and contour tracing in natural images tasks. Figure 1 shows the architectures of our models and their parameters. Training details are described in Section A.1. The model was compared with a feed-forward control with matching capacity. In the control, the CI layer was replaced by a block of sequentially arranged convolutional layers and did not use recurrence. The block contained an equivalent number of convolutional layers and each layer had the same configuration and parameters. Dropout layers with a dropout probability of 0.3 were added after every convolutional layer. This was necessary to prevent the control from over-fitting training data. For ease of notation we use CI layer to refer to this block as well and use the model type to distinguish between the two (model vs. control).

## 3  Experiments

### 3.1  Synthetic contour fragments

As a first task, we use stimuli typically used to study contour integration [1, 2]. Li et. al. [21] found that macaque monkeys progressively improved at detecting contours and had higher contour enhanced

Table 1: Synthetic contour fragment detection

| Name | Train | Validation | Test |
|------|-------|-----------|------|
| Model | 82.21 % | 79.79 % | 83.46 % |
| Control | 71.04 % | 72.72 % | 78.85 % |

Table 2: Contour tracing in natural images.

| Name | Train | Validation |
|------|-------|-----------|
| Model | $86.96 \pm 0.34$ % | $86.91 \pm 0.34$ % |
| Control | $76.53 \pm 0.76$ % | $76.41 \pm 0.76$ % |

V1 responses with experience on these stimuli. Hence, contour integration is learnable from these stimuli. Each stimulus consisted of several identical Gabor fragments that differed only in orientation. The orientations of a few adjacent (contour) fragments were aligned to form a smooth contour. The orientations of the remaining (background) fragments varied randomly. Models were tasked with identifying contour fragments. Embedded contours differed in their location, orientation, length $l_c$ (number of fragments), inter-fragment curvature $\beta$, and the Gabor fragment used in their construction. Example stimuli are shown in Figure 2A and B and Section A.2 describes how they were constructed. The dataset contained 64,000 training and 6,400 validation images. In its construction, 64 different Gabors, $l_c$ of 1, 3, 5, 7, 9 fragments and $\beta$ rotations of $0°, \pm15°$ were used. Gabor parameters were randomly selected with the only restriction that the resultant Gabor fragment appear as a well defined line segment. For $l_c = 1$, the label was set to all zeros. Contour integration requires inputs from outside the cRF and the model had to learn when not to apply enhancement gains as well. Equal number of images were generated for each condition.

**Effect of contour length and fragment spacing** Trained models were tested for consistency with behavioral and neurophysiological data with centrally located straight contours only (consistent with available neurophysiological data [2]). Behavioral performance was measured as the average intersection-over-union (IoU) of model predictions and labels. Neurophysiological responses were monitored at the output of the CI layer of centrally located neurons of each channel. For each channel, first, the optimal stimulus was found by monitoring which of the 64 Gabor fragments elicited the maximum response in the cRF. Next, to construct test stimuli, the first contour fragment was centered at the image center such that it was fully contained within the cRF of monitored neurons. Contours where extended in the preferred direction until the desired length and with the desired spacing. Finally, background fragments were added. Neurophysiological responses were quantified by the contour integration gain, $G(l_c, RCD) = \frac{\text{Output } l_c, RCD}{\text{Output } l_c=1, RCD=1}$, where the relative co-linear distance (RCD) [2] is the ratio of inter-fragment spacing to fragment length in pixels. The effects of contour length were analyzed using $l_c = 1, 3, 5, 7, 9$ fragments and a fixed spacing of RCD=1 (see Figure 2C). The effects of inter-fragment spacing were analyzed using RCD = [7, 8, 9, 10, 11, 12, 13, 14] / 7 and a fixed $l_c = 7$ fragments (see Figure 2D). For each condition, results were averaged across 50 different images.

## 3.2 Contour tracing in natural images

To investigate how contour integration is learnt in our natural viewing environment, we created a new task on natural images. For input images, we used the Barcelona Images for Perceptual Edge Detection (BIPED) dataset [20] as it focuses on all contours rather than only on object boundaries. Contour integration is a low-level phenomenon that occurs in shallow layers whereas object awareness typically develops in deeper layers. To create a stimulus, two smooth non-overlapping contours were randomly selected from an image's edges label. Next, two easily identifiable markers were placed on the contours. In some images, markers were placed to the same contour, while in others they were placed onto different contours. Markers were added to input images and the models never saw the selected contours. Finally, contours were fragmented by randomly puncturing the image with occlusion bubbles. Models had to report whether markers laid on the same smooth contour. An example stimulus is shown in Figure 3C. Stimulus construction is described in Section A.3. The train dataset contained 50,000 contours from BIPED train images while the validation dataset contained 5,000 contours from BIPED test images. Since the test dataset contains only 50 images, multiple contours per image were extracted. Equal probabilities were used for both classes.

**Effect of fragment spacing** After training, models were tested for behavioral and neurophysiological consistency using contours with connected end-points that were fragmented in an orderly manner. Different from training, bubbles were added along contours at calculated positions to fragment contours with fixed inter-fragment spacing. An example test stimulus is shown in Figure 3D and the procedure used to construct it is described in Section A.4. Contours were punctured with bubbles of sizes 7, 9, 11, 13, 15, 17 pixels, corresponding to fragment spacing of [7, 9, 11, 13, 15, 17]/7 RCD. Binary classification accuracy was used to measure behavioral performance. Neurophysiological responses were quantified by the contour integration gain for natural images,

$G_{NI}(rcd) = \frac{Output\ RCD}{Output\ RCD=1}$. Each channel of the CI layer was analyzed individually and results were averaged over 50 stimuli.

## 4 Results

### 4.1 Synthetic contour fragments

Mean IoU scores, averaged across 3 runs, are shown in Table 1. The model performed $\approx 7\%$ better than the control. Table 1 also shows mean IoU scores over all centrally located straight contours from our analysis of the effects of contour lengths (test). Both the model and the control found it easier to detect these contours with the model being $\approx 4\%$ better. While testing, we had an additional constraint of only considering neurons for whom the optimal stimulus was found (non-zero CI layer output for any single Gabor fragment in the cRF). Out of the 192 possible, 185 model and 46 control neurons met this criteria. Average IoU scores as contour length increases are shown in Figure 4A. Both the model and control excelled ($\geq 97\%$) at detecting the absence of contours. There were dips in performance for length 3 contours as they were the hardest to detect. For all other lengths, prediction accuracy increased with length.

Larger contrasts between the model and control was observed in neurophysiological gains. Figure 4B shows population average gains as contour lengths changed. For the model, gains increased monotonically with contour length. Control gains were more variable; gains dropped for length 3 contours but increased for other lengths. In general, model gains were higher than control gains for all lengths. Figure 4B also shows measured neurophysiological gains from Li et. al. [2]. Here, population average gains from the two monkeys used in their study were extracted using WebPlotDigitizer [22] and their weighted averages are plotted. The impact of contour length on measured and model gains was consistent. Figure 4C shows population average gains as the spacing between fragments was increased. Model gains decreased while control gains increased with spacing. The impact of spacing on measured and model gains was similar. For population average gains, we additionally removed neurons that were unresponsive to any contour condition (all zero gains) and those that had gains above 20 for any contour condition. Typically, these large gains were seen for neurons that had small responses to $l_c = 1, RCD = 1$ contours and small changes in the CI layer outputs significantly affected their gains. For the contour length investigation, we removed an additional 27 model and 34 control neurons while for fragment spacing we removed an additional 29 model and 36 control neurons. To get a better picture of overall trends, we plotted histograms of the gradients of linear fits to CI layer outputs vs. length and vs. spacing curves of all neurons for whom the optimal stimulus was found. Results of the model are shown in figure 4D and E while those of the control are shown in Figures 4F and G. Most model neurons were consistent with population average trends and showed positive slopes as contour lengths increased and negative slope when fragment spacing increased. The results of the model are consistent with observed neurophysiological trends while the control behaved differently. Remarkably, their behavioral predictions were comparable. The model and control appear to be employing different strategies to solve the task and only the model aligns with neurophysiology.

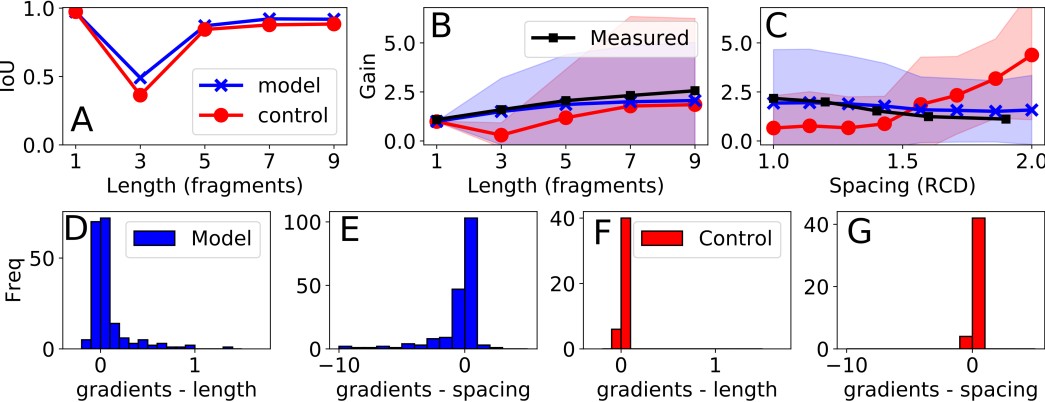

Figure 4: Synthetic contour fragments results. A, IoU vs. contour length. Behavioral classification accuracy increased with contour length. B and C, Population average gains vs. length and vs. fragment spacing. D and E, Gradients of linear fits of CI layer outputs vs. length and vs. spacing curves of individual channels of the model. F and G, similar plots as D and E but for the control. The model shows consistent trends with neurophysiological data while the control behaved differently.

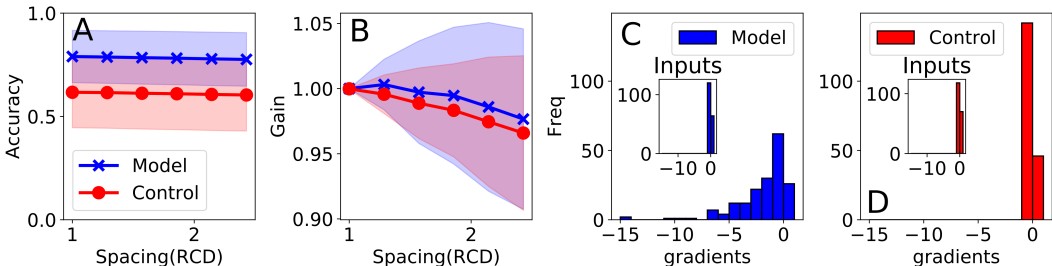

Figure 5: Contour tracing in natural images results. A, classification accuracy of the model (blue) and control (red) vs. fragment spacing. Model accuracy was significantly higher. B, population average gains ($G_{NI}$) vs. spacing. C and D, histograms of gradients of linear fits of CI layer outputs vs. spacing curves for individual channels. Gains decreased as spacing increased. Model is more sensitive to spacing between fragments. Insets show similar histograms but at the inputs of the CI layer.

## 4.2 Contour tracing in natural images

Classification accuracies, averages across 3 runs, are shown in Table 2. The model performed $\approx 11\%$ better than the control. When occlusion bubbles were added along contours (test data), the predictions of both the model and control dropped, even for the smallest bubbles. Accuracies as the spacing between contour fragments increased are shown in Figure 5A. For both the model and the control, accuracies dropped as spacing increased, consistent with observed behavioral properties. From the least to the most spacing configuration, model accuracy dropped by $\approx 3\%$ which was slightly more than that of the control, $\approx 1\%$. However, the drop in performance from the training task where bubbles were randomly located to when they we placed along the contours was significantly lower for the model (7%) compared to the control (14%) showing that model generalizes better. Figure 5B shows population averaged contour integration gains, $G_{NI}$, as the spacing between fragments increases. Both model and control gains dropped as spacing increased, consistent with observed neurophysiological trends. In contrast, where the output activations of individual neurons were compared, there was a larger difference between the model and control. For each channel, we did a linear fit to the output activation vs. fragment spacing curves and plotted the histogram of their gradients. Histograms for the model are shown in Figure 5C while those of the control are shown in Figure 5D. Model output activations dropped sharply as spacing increased (consistent with observed neurophysiological trends) while control outputs only slightly dropped.

## 5 Discussion

We found that brain-like contour integration can be learnt in artificial neural networks from high-level tasks. However, models need to be architecturally constrained. Moreover, it is important to validate models at multiple comparison points. From naturalistic scenes, contour integration can be learnt by tracing smooth natural contours. We plan to explore other more complex high-level tasks where such a capability would be useful. We also plan to quantitatively compare other potential mechanisms of contour integration. In particular, we are interested in the neurophysiological properties observed by Chen et. al. [23]. Here, it is proposed that contour integration is mediated by a recurrent loop involving V1 and V4 cortices.

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

# A    Appendix

## A.1    Training process

Models were trained using gradient descent with the ADAM [24] optimizer. For the synthetic contour fragments detection task, the starting learning rate was 3e-5, while for the contour tracing in natural images task it was 1e-4. Learning rates were dropped by a factor of 10 every 30 epochs. For loss we used Binary Cross Entropy plus a lateral connections weight constraint. To encourage sparse lateral connections we used L1 regularization loss with a inverse 2D Gaussian mask. The mask penalized weights that were far away from the center and encouraged them to be small. Regularization loss was weighted by 1e-5 before adding to BCE loss. The width of the Gaussian mask ($\sigma$) was set to 10 pixels. Models were trained for at least 50 epochs with a batch size of 32. All input images were 256×256 pixels. Each task was trained separately.

## A.2    Construction of synthetic contour fragments stimuli

The stimulus construction process was derived from [1]. First, an input image was sectioned into a grid of squares (full tiles) whose length was set to the pixel length of a fragment plus the desired inter-fragment spacing, $d_{full}$. The grid was aligned to map the center of the middle full tile to the image center. Each fragment was also a square the same size as feed-forward kernels. Input images were initialized with the channel-wise mean pixel value of the chosen Gabor fragment to blend in its edges. Second, a starting contour fragment was randomly placed in the image. Third, the location of the next contour fragment was determined by projecting a vector of length $d_{full} \pm {}^{d_{full}}/8$ and orientation equal to the previous fragment's orientation $\pm\beta$. The random direction of $\beta$ and distance jitter were added to prevent them appearing as cues. Fourth, a fragment was rotated by $\beta$ and added at this position. The third and fourth steps were repeated until $\lfloor l_c/2 \rfloor$ contour fragments were added to both ends of the starting fragment. Next, background fragments were added to all unoccupied full tiles. Background fragments were randomly rotated and positioned inside the larger full tiles. Lastly, a binary label of whether the center of a contour fragment was present was generated for each full tile. Input image size was fixed to 256×256 pixels, resulting in labels of size 19×19.

All training stimuli used a fixed inter-fragment spacing of RCD=1. In test stimuli, variable inter-fragment spacing was modeled by changing $d_{full}$ while keeping the fragment length constant.

## A.3    Construction of training stimuli for contour tracing in natural images

To construct a stimulus, a random smooth contour $C_1$ was extracted from an (image, edge map) pair in the BIPED dataset. Contours were extracted by selecting a starting edge pixel and incrementally adding contiguous edge pixels if they met a smoothness constraint that limited contour curvature to less than $\pi/4$ radians. One endpoint of $C_1$ was chosen as the position of the first marker, $M_1$. Next, a second edge pixel that did not lie on $C_1$ was randomly selected. To ensure connected and not connected stimuli had similar separation distances, the selection process used a non-uniform probability distribution to favor edge pixels that were equidistant with the unselected endpoint of $C_1$. A second contour, $C_2$, was extended from the second edge pixel. If $C_2$ overlapped with $C_1$, the process was repeated until a non-overlapping pair of contours was found. The location of the second marker $M_2$ was determined by the type of stimulus. For connected stimuli, $M_2$ was the opposite end of $C_1$, while for not connected stimuli, one of the endpoints of $C_2$ was chosen. Once marker positions were determined, markers were added to the corresponding input image. The marker was a bulls-eye of alternating red and blue concentric circles (see Figure 3B).

To fragment contours, occlusion bubbles were randomly added to input images. Following [25], 2D Gaussian bubbles were used to reduce the impact of bubble edges. Occluded parts were replaced by channel means. Each image was punctured by 300 bubbles of various sizes. Bubble sizes were specified by the full-width-half-magnitude (FWHM) of the chosen 2D Gaussian. Bubbles were blended into the image over a square area defined by 2×FWHM of a bubble. The location and size of bubbles was specified in a bubble mask which was blended into the image using, $img_{punc} = mask \times img + (1 - mask) \times mean_{ch}$. Bubbles were allowed to overlap and a different mask was used for each image (see Figure 3C).

## A.4 Construction of test stimuli for contour tracing in natural images

In the synthetic dataset, inputs were altered to find the optimal stimulus of monitored neurons. However, for natural images, inputs cannot be similarly changed. Therefore, a new procedure was devised to find optimal stimuli. To find optimal stimuli in natural images, multiple unoccluded connected contours were presented to the model (Figure 3B). New random contours were selected from the BIPED train dataset. We used the train dataset as opposed to the test dataset as it contained more images and a larger variety of contours. For each image, the position of the most active neuron of each channel in the CI layer was found. If it was within 3 pixels (the same as the stride length of the subsequent convolutional layer) of the contour, the image as well as the position of most active neuron were stored. The process was repeated over 20,000 contours and the top 50 (contour, most active neuron) pairs were retained for each channel.

Given the optimal stimuli for each channel, each input contour was fragmented by inserted occlusion bubbles at specific positions along the contour. Multiple bubble sizes were used to fragment contours with different inter-fragment spacing. A fixed fragment length of 7 pixels, the same size as the cRF, was used. To ensure the cRF of the most active neuron was not affected, we found the position of the closest point on the contour. Bubbles were then inserted along the contour at offsets of $\pm \frac{(l_{frag}+l_{bubble})}{2}, \pm \frac{3(l_{frag}+l_{bubble})}{2}, \pm \frac{5(l_{frag}+l_{bubble})}{2}, \ldots$ until the ends of the contour. Finally, we restricted the blending-in area of bubbles to FWHM pixels to ensure visible contour fragments were unaffected. This gave bubbles a square profile but as can be seen in the inset plots of Figure 5C and D), it did not significantly impact contour integration layer inputs.

