# OpenReview forum: "Task-Driven Learning of Contour Integration Responses in a V1 Model"
_NeurIPS.cc/2020/Workshop/SVRHM — SVRHM@NeurIPS Oral_

### Official Review · AnonReviewer3 · 2020-10-24
**The author(s) implemented an artificial neural network based on V1 architecture for contour integration tasks. They evaluated the effects of contour length and spacing between contour fragments for both synthetic stimuli and natural images. This contributes to understanding how we can build biologically plausible networks to simulate partial brain mechanisms.**

**Rating:** 8
**Confidence:** 4

**Review:**

Model:

Pros:
1. clear explanations of the model and how it's different from most of the RNN models
2. biologically plausible based on V1 neuron structures

Cons:
1. figure 3. It is not clear how figure 3D is different from figure 3B for testing the impact of fragment spacing.

Experiment:

Pros:
1. used both well-manipulated synthetic stimuli and natural images for training and testing
2. clear descriptions on how to create stimuli and stimuli conditions in both tasks

Cons:
1. There is some inconsistency between the fragments in synthetic contour fragments and natural image contour fragments. The fragments in synthetic stimuli have clear endpoints at the end of each fragment. While the contours in the natural images were punctured using bubbles to create fragments, which led to smoother endpoints instead of clear-cut endpoints.

---

### Official Review · AnonReviewer1 · 2020-10-28
**A very interesting neuroanatomical block for contour integration improves performance in relevant tasks in shallow neural networks**

**Rating:** 9
**Confidence:** 4

**Review:**

The authors present a neuroanatomically realistic model of contour integration and show that it improves performance in contour fragment detection and contour tracing in natural images tasks versus a control model. Furthermore the authors perform a series of analyses of the responses of units in the model and qualitatively compare them to known properties of primate V1 neurons.

Strengths:
+ Model is very interesting and the contour integration block could potentially be used to improve object recognition systems
+ Paper is clear and very well written
+ Analyses are very thorough and properly carried out
+ Work is properly contextualized with a very good introduction summarizing the relevant literature

Weaknesses:
- While the analyses of the neuronal responses are interesting, unfortunately, the authors fall short of quantifying exactly how well they approximate the observed phenomena in primate primary visual cortex

Questions:
L160-162:
It is not clear the criteria for inclusion of the neurons. Only neurons that responded to a single optimally oriented fragment in the cRF were considered. Does that mean that only neurons that were orientation selective, that is, that respond stronger for one orientation versus others? Why an activation greater than 0.01? What are the range of activations of neurons? Why is there such a big difference in the number of selected neurons between the model and control?

L254-263:
In the training process section the training procedures for the two tasks (synthetic contour fragments detection, and contour tracing in natural images) are described separately. However both classifiers share the same contour integration block and therefore it is not clear in which order of the tasks it was trained. Was it trained on one and then on the other? Or was it trained on both at the same time by alternating epochs on the two tasks. Given that the learning rate of the synthetic contour fragments task was higher, doesn’t this bias the contour block to better perform on this task?

Figure 4B and C:
Does the population average contain all the neurons or only the neurons that pass the criteria described in L160-162? Does this mean that in the control, when there are 3 segments or more responses of the neurons in the center are completely silenced? If so could the authors comment on it?

Figure 4A:
Could the authors comment on why the control model does better for the most challenging stimulus (3 fragments)?

Figure 4D and E:
This suggests that the majority of the neurons is not modulated by the surround and only a small fraction of the neurons are enhanced by the presence of co-aligned fragments. Could the authors comment on this? How does it relate to empirical data from the primate V1?

Minor corrections:
L80 - Typo: ResNet50
Figure 1 - Typo: “sieze” -> size
L109 - “IoU” is not defined
L158 - “Figures 6D and E” -> Figures 4D and E
L160 - “Figures 6F and G” -> Figures 4F and G

---

### Official Review · AnonReviewer2 · 2020-10-29
**Well motivated, thorough model with experiments and results in progress**

**Rating:** 6
**Confidence:** 4

**Review:**

Contour integration in V1 happens when stimuli outside the classical receptive field (cRF) modulates neural responses, in particular responses are enhanced if preferred stimulus in the cRF is part of a larger contour. In this paper, the authors propose a model of contour integration embedded in a network trained to do two tasks: “contour fragment detection” on synthetic stimuli , and “contour tracing” using natural images.  Their model has an excitation-inhibition layer that draws inspiration from previous biological models. The authors claim that their trained models exhibit known behavioral  and neurophysiological effects: 1) easier discrimination of longer contours, 2) gain in CI layer outputs  with longer contours, and 3) decrease in CI layer outputs as spacing between fragments of a contour increases.

Overall, the paper addresses an important question that adds to a growing body of work modeling CI, is well motivated, and details the modeling approach thoroughly. However, the experiments and results still constitute work in progress that lacks substance to support any significant implications for computer vision, or visual neural modeling.

I here outline a few comments/suggestions that may help the current work increase its significance:

* How does the performance of their model compare to current SOTA in contour fragment detection? Although the authors say that their goal is not to achieve SOTA, it would help to include other better performing models in both the synthetic and natural image tasks.
* I am concerned that the “task-driven” objectives that the authors use is too close to the properties that the authors expect to match with the brain. It feels intuitive to me, that if one trains a net on a contour fragments detection task, then the increase accuracy with contour length is a natural result for that model. This is further supported by the fact that the control feedforward model had similar behavioral classification accuracy (Fig 4A).
I would encourage the authors to embed their CI layer in networks trained on a more high level task (e.g. object recognition). Presumably, contour integration is a useful feature since our visual system does it. If the “brain-like”properties ( i.e higher classification accuracy with increase lengths, output gains with longer fragments, output suppression with spaced fragments) emerge by reading out from the early CI layer, this would support a lot more their message.
*  The increase in gain in response as a fraction of length of the fragments seems very variable, with a very slight increase in avg. response. How variable, and by how much do recorded neural responses change in the brain with increasing length (fragments)? The same goes for the slight output decrease as a function of spacing.
* It was unclear to me for results in Fig 4B and 4C what the output of the control model was (what layer / feature maps). Couldn’t one find a direction in the output feature space of the feed-forward control that exhibit these neurophys. properties?
* I found the contour tracing task on natural images that the authors propose interesting and creative. However, I think that a few more variations in the occlusion settings for training and testing could control for unwanted spurious effects/shortcuts solving the task. For example I would change the hue of the occlusion bubbles (e.g. to match the average color of the occluded image patch)
* I would follow the authors’ original motivation and test for increase in robustness to corruptions or adversarial perturbations. when using a CI layer embedded in an image classification network.
* Finally, can their task-driven model be used to predict recorded neural responses to similar stimuli? I’m not sure if the recordings from Li et al are available, but this a system identification approach using the CI layers' feature space would surely provide stronger evidence for brain-like computations.

---

### Decision · Program_Chairs · 2020-11-02

Accept (Oral)